# Decoupled Human-Object Interaction Inference Addressing Architectural Order Dependency in the Query-based Model

## Abstract

Human–object interaction (HOI) inference is a crucial component of end-to-end HOI detection, responsible for predicting the interactions between humans and objects in an image. While query-based detectors have achieved state-of-the-art performance in HOI detection, their interaction inference modules are typically tightly coupled with the detection pipeline, hindering independent evaluation and optimization. Recent research suggests that decoupling this module can improve overall detection, yet its standalone effectiveness remains underexplored. To this end, we introduce a dedicated evaluation framework for isolated HOI inference modules and identifies two key factors limiting current performance: architectural order dependency and dataset impurities. To address these issues, we propose a novel interaction inference model that removes self-attention from the decoder and introduce dataset refinement strategies, including verb clustering and redundant bounding-box unification. Extensive experiments on multiple benchmarks demonstrate that our approach surpasses existing inference modules by an average of $20\%$ $m$AP, confirming its effectiveness and robustness, and the optimization of the decoupled interaction inference model further improves the end-to-end model. Code and data are publicly available at Decoupled-HOII-AAOD.

## 1 Introduction

Human–object interaction (HOI) detection is a fundamental problem in computer vision, requiring both the spatial localization of human–object entities and the classification of their interactions. As a cornerstone of visual scene understanding, HOI detection has attracted increasing attention due to its relevance in applications such as assistive robotics, visual surveillance, and video analysis (Bemelmans et al., 2012; Bolme et al., 2010; Dee & Velastin, 2008; Feichtenhofer et al., 2017).

The task typically involves three sequential components: object detection, human–object pair association, and interaction inference (Figure 1). These sub-problems are rarely solved in isolation. Early multi-stream approaches, such as HO-RCNN (Chao et al., 2018b), InteractNet (Gkioxari et al.,

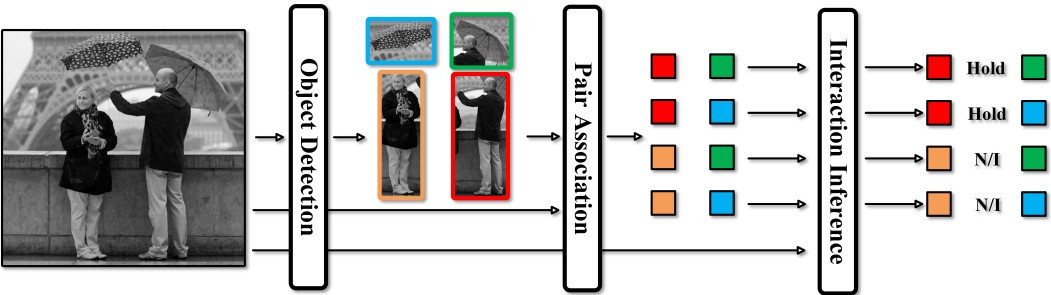

Figure 1: The three-stage paradigm for the end-to-end HOI detection. The image is presented from the HICO-DET dataset (Chao et al., 2018a). N/I stands for no interaction.

2018), and PD-Net (Zhong et al., 2020), as well as graph-based models including GPNN (Qi et al., 2018), RPNN (Zhou & Chi, 2019), and SCG (Frederic Z. Zhang & Gould, 2021), generally first detect objects, followed by joint reasoning over pair associations and interactions.

More recently, inspired by the success of DETR (Carion et al., 2020) in object detection, query-based HOI detectors (e.g., HOI-Trans (Zou et al., 2021), QPIC (Tamura et al., 2021), GEN-VLKT (Liao et al., 2022), HOICLIP (Ning et al., 2023), RLIP (Yuan et al., 2022)) have emerged. These methods unify object detection and pair association into a single query-based framework and employ various architectural designs to address challenges in interaction inference, including zero-shot generalization and long-tail distribution issues. However, by design, query-based models tightly couple the final-stage interaction inference module with the preceding detection pipeline, preventing its independent evaluation and optimization.

This paper focuses on the final stage of human–object interaction (HOI) detection—the interaction inference task—with particular attention to its role within query-based models, which have demonstrated superior end-to-end performance compared to other architectures. Despite their success, interaction inference modules in these frameworks remain inherently coupled to the detection pipeline, limiting their independent evaluation and optimization. RLIPv2 (Yuan et al., 2023) introduced an interaction inference module, R-Tagger, structurally identical to the end-to-end detector, and showed that the end-to-end model can benefit from pseudo-labels generated by such a module.

Building on these observations, this paper argues that decoupling interaction inference modules offers several advantages that warrant independent analysis:

1. Upper-bound performance estimation: The performance of a standalone interaction inference model can approximate the upper bound of structurally identical end-to-end detectors, providing valuable insights for guiding future improvements in integrated models.

2. Flexible integration with other components: Decoupled inference models can be cascaded with object detection and image captioning modules to construct end-to-end HOI detection systems capable of generating pseudo-labels for unlabeled images, with label quality further enhanced when partial annotations are available.

3. Improved model initialization: High-quality pre-trained weights obtained from standalone inference models can be transferred to isomorphic end-to-end architectures, facilitating faster convergence and improved training stability.

However, preliminary experiments show that **R-Tagger falls short of expectations**, exhibiting substantial misattribution in pseudo-label generation. To address these issues, this paper adopts a systematic methodology that defines objective evaluation metrics and introduces improvements from both the model and data perspectives. The main contributions are summarized as follows:

1. **Architectural analysis and refinement:** We investigate the order dependencies within the R-Tagger architecture, analyze their impact on performance stability, and propose targeted architectural modifications to mitigate these limitations. The refinement in the decoupled HOI inference model can also be incorporated into the end-to-end HOI detection model to boost the performance.

2. **Novel evaluation metrics:** We introduce evaluation metrics specifically designed for HOI inference models, which effectively assess model performance on both positive and negative samples.

3. **Dataset cleansing and open-source resources:** We develop dataset refinement techniques, including redundant bounding-box unification and verb clustering, to resolve ambiguities during training and evaluation. The processed dataset is open-sourced to promote reproducible research and accelerate advancements in the HOI detection community.

## 2 RELATED WORK

### 2.1 QUERY-BASED MODEL

The query-based paradigm, first introduced by DETR (Carion et al., 2020), has significantly advanced object detection and its related fields, including HOI detection, by leveraging a set prediction

framework. A comparison between the evolution of query-based models in HOI detection and the development of DETR models reveals that the latter has consistently guided progress in the former. Enhancements to DETR backbones, such as Deformable-DETR (Zhu et al., 2021), DAB-DETR (Liu et al., 2022), and $\mathcal{H}$-DETR (Jia et al., 2023), have similarly demonstrated performance gains when applied to HOI detection tasks (Yuan et al., 2022; 2023; Zhang et al., 2023). Building on these insights, this paper revisits the query-based paradigm and adapts it to the interaction inference task by explicitly addressing the issue of architectural order dependency.

## 2.2 HUMAN-OBJECT INTERACTION INFERENCE

Research on the human–object interaction (HOI) inference task has received considerably less attention than work on end-to-end HOI detection. Due to the virtually limitless diversity of human–object relationships, existing HOI datasets typically provide annotations only for a subset of relation categories, which limits model generalization. To address this challenge, recent studies (Ning et al., 2023; Liao et al., 2022; Wu et al., 2023) have explored knowledge transfer from vision–language models (VLMs) to improve the reasoning capability of HOI inference modules. However, unlike conventional modules, VLM-based reasoning relies heavily on external priors, necessitating careful assessment of their influence. For instance, Park et al. (2025) reports that CLIP fails to handle negation, a critical limitation for HOI models that depend on CLIP priors, since "no interaction" is among the most important relationship categories.

In this work, we argue that evaluating decoupled interaction inference models is essential for accurately assessing their reasoning ability and for understanding the performance of end-to-end HOI detectors. To this end, we construct a dedicated evaluation framework that enables a more precise analysis of interaction inference in isolation. Furthermore, we leverage this framework to systematically improve the design of R-Tagger, achieving notable performance gains.

## 3 DATASET

The original R-Tagger was trained on the Visual Genome dataset (Krishna et al., 2017). This paper argues that generalization performance cannot be adequately reflected through a single dataset and thus introduces two widely used HOI datasets—HICO-DET (Chao et al., 2018a) and V-COCO (Gupta & Malik, 2015). Labels in the HICO-DET and V-COCO datasets consist of three components: object classes, object bounding boxes, and relationships between human-object (H–O) pairs. While the labels in the VG dataset is more complex, this paper only focuses on the same three components of the labels.

### 3.1 REDUNDANT BOUNDING BOXES UNIFICATION

Both the HICO-DET and VG datasets suffer from superfluous detection annotations on the same objects, i.e. redundant bounding boxes. Since the training pipeline of R-Tagger takes unlabeld H–O pairs as negative samples, redundant bounding boxes on the same object can lead to significant ambiguity. Therefore, a unification algorithm is necessary to cleanse these boxes.

Figure 2 visualizes the proposed method. The proposed unification algorithm first clusters all the boxes, with each cluster corresponding to a single identical object, and then unifies the labels in each

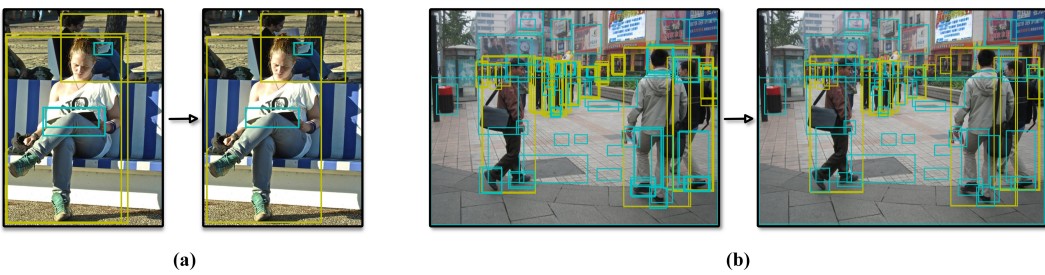

(a)                                                                          (b)

Figure 2: The proposed unification algorithm utilized on (a) HICO-DET and (b) VG dataset.

cluster. The goal of clustering is to sort all the boxes according to their actual bounding objects. If a cluster contains more than one object, it indicates the presence of redundant bounding boxes within that cluster. Note that due to the subsequent processing approach, the cluster center in this context is not important and can be chosen arbitrarily from the cluster.

The proposed algorithm iterates through all existing clusters for each bounding box, attempting to determine which current cluster the box belongs to. A bounding box is considered redundant to a cluster of objects if and only if: **1.** The object category is the same with all the objects in the cluster; **2.** The bounding box has Intersection over Union (IOU) values of at least $t_l$ with all the boxes in the cluster; and **3.** The IOU of the bounding box and at least one in the cluster exceeds $t_h$. The bounding box is assigned to a new cluster if it cannot be added to any existing cluster. In practice, this paper takes $t_h = 0.5$ and $t_l = 0.3$.

After the cluster map is obtained, there are two methods to unify the labels within each cluster: 1. **Remove** the other objects, leaving their relationship labels to the cluster center. 2. **Boardcast** all the relationship labels from each object to the other objects in the cluster.

This paper uses the subscript r (remove) to denote the unification method 1 and subscript b (boardcast) to denote the unification method 2. We carefully compare the two methods by conducting an experiment, as detailed in Table 2 and section 5.1. The results indicate that the latter method performs better.

### 3.2 VERB CLUSTERING

This section aims to further cleanse the relationship labels (i.e., verb labels) to eliminate ambiguity.

The motivation for verb clustering stems from the complex verb distribution in the VG dataset, where numerous variants of the same verb prototype exist, including various inflected forms, phrasal constructions, and misspelled versions. Preliminary experiments reveal that a considerable portion of text encoders, including the RoBERTa-base model utilized by R-Tagger, struggle to encode these variations sufficiently close to one another, as they are trained within sentence- or paragraph-level contexts. While certain text encoders, such as the all-MiniLM-L6-v2 model[1] (Wang et al., 2020) mentioned below, are capable of handling such tasks, this study does not focus on the encoder performance within specific verb collections, and opt to cluster verbs sharing identical prototypes prior to training and to represent them uniformly using their base forms.

Before the verb clustering, we need to deal with some special cases. The first thing is about the representation of the "no interaction" label, which is explicitly present in the HICO-DET dataset but absent in the other datasets. This paper uniformly represents the "no interaction" label as an all-zero label across the global relationship vocabulary, as the interaction between a human-object pair labeled as "no interaction" may be represented by a verb that is not included in the relationship vocabulary. Another issue arises in the V-COCO dataset, which contains interactions without objects. This paper excludes these relationships from both the training and evaluation processes, deferring this type of interaction to downstream fine-tuning on the V-COCO dataset.

Given the high quality of verbs in both the HICO-DET and V-COCO datasets, the clustering algorithm treats the words and phrases therein as cluster centers, totaling 128. Subsequently, the all-MiniLM-L6-v2 text encoder is utilized to compute cosine similarity between each relationship label in the VG dataset and these 128 cluster centers. The cluster center with the highest similarity score is designated as the cluster proposal for each relationship label. Then, a threshold is chosen through the precision-recall analysis experiment based on manually annotated samples, whereby labels exceeding this threshold are assigned to their corresponding clusters.

Excluding images without human, the VG dataset has 8063 relationship labels, including 1362 kinds of words, 3201 kinds of two-word phrases, and 3500 kinds of longer phrases. Distinct thresholds are selected for each of these three categories. In practice, 396 words, 1758 two-word phrases, and 973 longer phrases are recalled. Finally, the clustering results undergo manual verification, supplementary clustering is applied to the verb "play", and prepositions including "on", "at", "with", and "under" are filtered out. In summary, 2848 kinds of relationship labels in the VG datasets are clustered into 129 categories. For clarity, we denote the VG dataset after verb clustering as VG$'$

---

[1]All-MiniLM-L6-v2 is a fine-tuned MiniLM on sentence-transformers (Reimers & Gurevych, 2019).

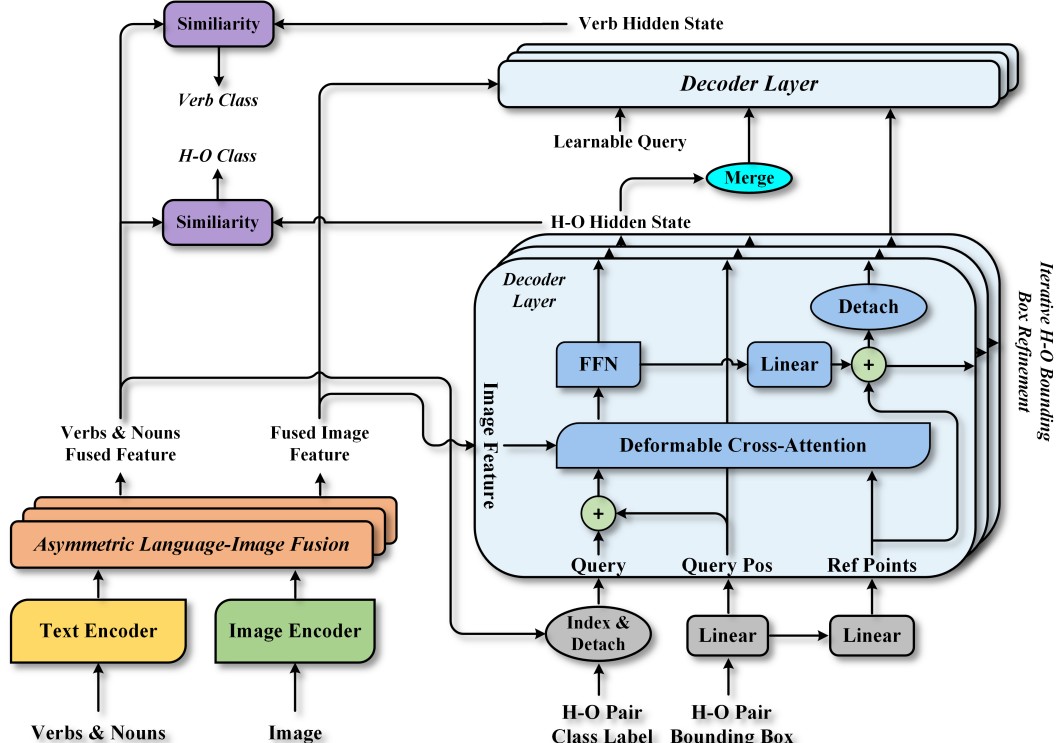

Figure 3: The overview of the proposed model. The asymmetric language-image fusion module is identical to that in Yuan et al. (2023). In practice, this paper utilizes ResNet-50 (He et al., 2015) for the image encoder and RoBERTa-base (Liu et al., 2019) for the text encoder.

throughout this paper. And due to the absence of the testset, we randomly splits the VG$'$ dataset into training and validation sets with a 9:1 ratio.

### 3.3 METRIC

As the interaction inference model works on detection results, the minimum input unit of the system consists of an image, a human-object pair with associated detection metadata, and a vocabulary of relationships to be inferred. For each human-object pair, the interaction inference model outputs the confidence for each verb. Note that the HICO-DET dataset restricts the object categories that can interact with each verb, but this paper does not employ this constraint in the evaluation.

In this paper, inference models are evaluated using two metrics: mean average precision over labeled pairs ($m\text{AP}_l$) and mean average precision over all pairs ($m\text{AP}_a$), both calculated by averaging the AP values across all relation categories. The metric $m\text{AP}_l$ exclusively assesses the model's reasoning ability on labeled human-object pairs, while $m\text{AP}_a$ further evaluates the model's performance on irrelevant H–O pairs by assigning unlabeled pairs as negative samples.

## 4 STUDY ON ARCHITECTURAL ORDER DEPENDENCY

We first begin by briefly outlining the architecture of R-Tagger (Yuan et al., 2023), then analyze the order dependencies embedded within its design, evaluate their impact on performance, and finally propose targeted solutions to address these issues.

R-Tagger adopts an encoder–decoder architecture. It takes as input an image, a vocabulary consisting of object classes (nouns) and relationships (verbs), as well as a batch of $N_p$ H–O pairs with their bounding boxes and class labels. The image encoder extracts visual features, while the text encoder encodes the noun and verb embeddings. These features are fused through an Asymmet-

Table 1: Performance of R-Tagger fine-tuned on the V-COCO, HICO-DET$_b$, and VG$'_b$ datasets under various configurations, measured on the HICO-DET$_b$ testset.

| Queries input | Vocabulary input | $m\text{AP}_l$ | $m\text{AP}_a$ |
|---|---|---|---|
| Each H–O pair once, labeled pairs first | Global verbs | 51.7% | 47.8% |
| Each H–O pair twice, labeled pairs first | Global verbs | 47.4% | 43.3% |
| Each H–O pair once, shuffle queries | Global verbs | 42.6% | 39.8% |
| Each H–O pair once, shuffle queries | Sample to 100 verbs | 42.7% | 39.4% |
| Each H–O pair once, shuffle queries | Sample to 90 verbs | 42.7% | 39.5% |
| Each H–O pair once, shuffle queries | Sample to 80 verbs | 42.3% | 39.9% |

ric Language–Image Fusion module, where the term asymmetric reflects the different depths of the image and text encoding layers, and the fusion is performed using gated cross-attention.

Structurally identical to the end-to-end detector RLIPv2, R-Tagger employs a two-stage deformable decoder (Zhu et al., 2021). In the end-to-end model, the first-stage decoder applies self-attention to $N_p$ learnable query pairs ($2N_p$ queries in total) and performs deformable cross-attention with fused image features to generate paired H–O hidden states. Object categories are predicted by computing similarities with fused noun features, while bounding boxes are refined iteratively using the same mechanism as Deformable DETR.

In the decoupled interaction inference setting, R-Tagger takes detection metadata of H–O pairs as input queries, effectively modeling the entire first-stage decoder as a reconstruction module. The second-stage decoder then processes $N_p$ learnable queries, producing $N_p$ verb hidden states. At each decoder layer, the paired H–O hidden states are merged and added to the query inputs, and the relationships for each H–O pair are inferred by computing similarities with the fused verb features.

Ideally, for a given image, the confidence score predicted by the interaction inference model for a specific relation in a human–object pair should be **order-invariant**—that is, it should remain unaffected by the composition or ordering of the input set. However, the architecture of **R-Tagger** deviates from this ideal due to three structural factors:

**Self-attention in the decoder:** The self-attention layers make the outputs dependent on the set of query inputs, as each query output is computed as

$$q'_i = \text{softmax}\left(\frac{q_i Q^T}{d}\right) Q,$$

where interactions among all queries influence the result. Consequently, the representation for an H–O pair becomes context-dependent; it is inevitably modulated by all other pairs concurrently present in the input set.

**Learnable queries in the second-stage decoder:** The use of distinct learnable queries, which are directly added to the outputs of the first-stage decoder, introduces sensitivity to the order of input queries, as permutations alter the corresponding query embeddings.

**Bi-directional cross-attention in the language–image fusion module:** This mechanism renders the outputs sensitive to the noun and verb vocabulary, since changes in the vocabulary modify the attention map and, consequently, the fused representations.

During training, labeled H–O pairs are consistently placed at the beginning of the input list; therefore, the model's performance under this configuration is regarded as the **ideal baseline**. As shown in Table 1, three sets of control groups are conducted to quantify the impact of the three factors:

**Factor 1 (Self-attention sensitivity):** To assess the effect of self-attention, we duplicated all H–O pairs while retaining labeled pairs at the beginning. Ideally, the model should assign identical confidence scores to the duplicated entries. However, experiments reveal a 4% drop in both metrics, indicating that factor 1 introduces significant instability into the model's predictions.

**Factor 2 (Order sensitivity of learnable queries):** To evaluate order sensitivity, we shuffled the input list of H–O pairs. This setup reflects realistic conditions where the order of pairs is unknown. The results show a 9.1% drop in $m\text{AP}_l$ and a 8.0% drop in $m\text{AP}_a$ compared to the baseline, confirming that factor 2 severely affects performance stability.

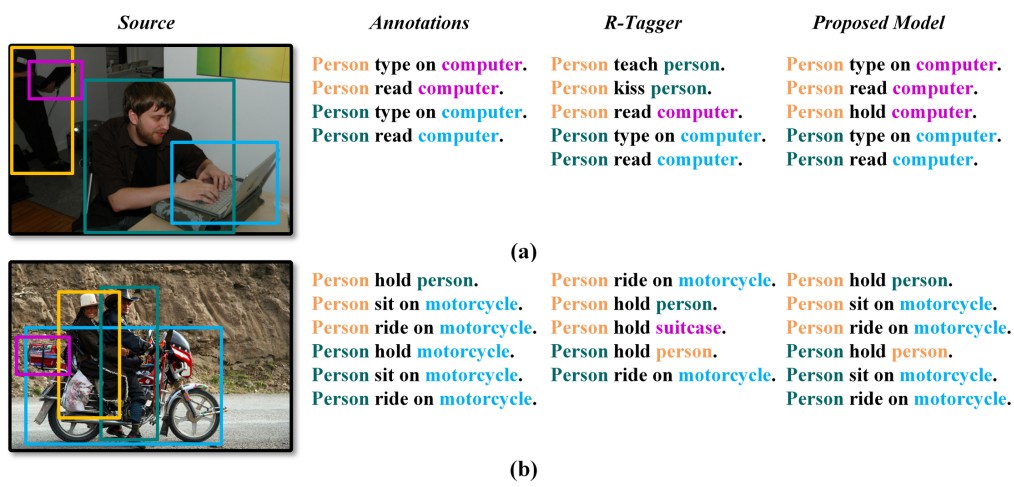

|  | *Source* | *Annotations* | *R-Tagger* | *Proposed Model* |

Figure 4: Qualitative improvements from our method on (a) HICO-DET and (b) V-COCO datasets.

**Factor 3 (Vocabulary sensitivity in fusion):** To analyze vocabulary dependency, we introduced three variants by expanding the verb set to 100, 90, and 80 verbs (sampled by frequency) in addition to the labeled verbs. The results exhibit only minor fluctuations, suggesting that the negative impact of factor 3 is limited. This finding aligns with the generally demonstrated robustness of vision–language fusion modules in detection models (Liu et al., 2024; Cheng et al., 2024).

As shown in Figure 3, this paper proposes removing the self-attention layers in the decoder and replacing the learnable queries of the second-stage decoder with a broadcast form derived from a single learnable query, thereby addressing the architectural order dependency of the decoder.

## 5 EXPERIMENTS

The training pipeline of the proposed model is slightly modified but fundamentally aligned with that of R-Tagger. In addition to human-object pairs with relationship labels, unlabeled pairs are also used as negative samples during training. The vocabulary input comprises all relationship categories in the training datasets, as well as the supersampled object classes. Specifically, in addition to the object classes present in the current samples, additional object classes are sampled based on their frequency in the training datasets. This paper also utilizes the identical denoising training, introducing noises to the first-stage decoder to better converge the reconstruction module.

The overall loss $\mathcal{L}$ consists of the loss from the first-stage decoder (reconstruction loss $\mathcal{L}_{\text{rec}}$) and the loss from the second-stage (relationship) decoder $\mathcal{L}_{\text{rel}}$:

$$\mathcal{L} = \mathcal{L}_{\text{rec}} + \mathcal{L}_{\text{rel}} \tag{1}$$

$$\mathcal{L}_{\text{rec}} = \lambda_1 \mathcal{L}_{l1} + \lambda_2 \mathcal{L}_{\text{GIoU}} + \lambda_3 \left( \mathcal{L}_h + \mathcal{L}_o \right) \tag{2}$$

$$\mathcal{L}_{\text{rel}} = \lambda_4 \sum_s \theta_s \mathcal{L}_{r,s}, \ s \in \{\text{labeled}, \text{unlabeled}\} \tag{3}$$

where $\mathcal{L}_{l1}$ and $\mathcal{L}_{\text{GIoU}}$ denote the $\ell_1$ loss and the GIoU loss (Rezatofighi et al., 2019) for box regression, $\mathcal{L}_h$ and $\mathcal{L}_o$ denote the cross-entropy loss for human and object classes, $\mathcal{L}_r$ denotes the focal loss (Lin et al., 2017) for relations, and the subscript $s$ denotes the subset of labeled pairs or unlabeled pairs. In practice, $\lambda_1$, $\lambda_2$, $\lambda_3$, and $\lambda_4$ are set to 2.5, 1, 1, and 1 respectively as fixed weight over different tasks; $\theta_{\text{labeled}}$ and $\theta_{\text{unlabeled}}$ are set to 1 and 24.

Given the additional datasets and based on preliminary experiments validating convergence adequacy, this paper adopts a training configuration with longer epochs for all the fine-tuned models unless otherwise specified: the AdamW optimizer (Loshchilov & Hutter, 2019) with an initial learning rate of 2e-4, learning rate decay by a factor of 10 every 15 epochs, and training for 33 epochs.

Table 2: The performance of the proposed model fine-tuned on different unification methods and evaluated on the testset of the HICO-DET$_b$ dataset.

| Unification mehod | $m\text{AP}_l$ |
|---|---|
| Not unify | 32.1% |
| Remove | 39.8% |
| Boardcast | 52.4% |

Following the R-Tagger configuration, the image and text encoders use learning rates that are one-tenth of those applied to other model components. All experiments are conducted with a batch-size of 2 across $8\times$RTX 3090 GPUs.

## 5.1 STUDY ON THE UNIFICATION METHODS

This section proposes to study on the better way to unify the clustered boxes. As mentioned above, the redundant bounding boxes unification method is applied to the HICO-DET dataset and the VG dataset. Since non-exhaustive annotations and annotation errors are much more prevalent in the VG dataset, experiments are conducted on the HICO-DET dataset, and the results are used to guide our approach to the VG dataset.

Models with pretrained weights from R-Tagger are fine-tuned on various unified datasets and evaluated on the testset of the HICO-DET$_b$ dataset. As shown in Table 2, experiment results demonstrate that the broadcast-based unification method performs better.

## 5.2 PERFORMANCE OF THE PROPOSED MODEL

In this section, we propose evaluations on the proposed method that addresses architectural order dependency in the query-based model. The $m\text{AP}_a$ metric is excluded specifically in the VG dataset due to the presence of non-exhaustive annotations and annotation errors.

As shown in Table 3, the experimental results demonstrate that the proposed model addressing architectural order dependency achieves significant improvements over R-Tagger across all datasets on both the $m\text{AP}$ value and the drop in $m\text{AP}_a$ compared to $m\text{AP}_l$. Specifically, the latter demonstrates the proposed model's improved performance in reasoning with negative samples. Figure 4 shows the improvements of the proposed model on both the labeled and unlabeled samples across the HICO-DET and V-COCO datasets. Note that due to the severe annotation errors and missing annotations in the VG dataset, even after the redundant bounding boxes unification, only a small portion of these issues can be resolved, resulting in low model performance on the VG dataset.

Given its demonstrated effectiveness in RTagger, we extend the proposed method to the structurally identical end-to-end model RLIPv2. Experiments demonstrate an 1% improvement in $m\text{AP}$ performance on the HICO-DET dataset and a 0.3% improvement on the V-COCO dataset. In further comparison with state-of-the-art query-based end-to-end HOI detection models, our work offers insights into the performance upper bounds achievable by this architecture on relevant datasets.

## 5.3 ABLATION STUDY

Although numerous ablation studies on R-Tagger have been conducted in Yuan et al. (2023), this section further proposes ablation studies on two key yet potentially controversial components.

Firstly, concerning that the output of the first-stage decoder can be directly transformed into equivalent input representations, it is reasonable to question whether this decoder could be bypassed, with the second-stage decoder input constructed directly from the original input. To comprehensively investigate this hypothesis, four control groups were established for comparative analysis:
**Control group 1:** Follow the proposed model, except without using $\mathcal{L}_{\text{rec}}$ in calculating the loss.
**Control group 2:** Remove the first-stage decoder and directly concatenating its input to the output.
**Control group 3:** Based on 2, double the number of layers in the second-stage decoder.
**Control group 4:** Based on 2, use learnable queries instead of indexed noun features.

Table 3: Evaluation results of the proposed Addressing Architectural Order Dependency (AAOD) method compared with R-Tagger and current remarkable query-based end-to-end HOI detection models, all utilizing a ResNet-50 backbone. R-Tagger' denotes R-Tagger fine-tuned on the HICO-DET$_b$, V-COCO, and VG$_b'$ datasets.

| Model | HICO-DET$_b$ testset | | V-COCO testset | | VG$_b'$ valset |
| | $mAP_l$ | $mAP_a$ | $mAP_l$ | $mAP_a$ | $mAP_l$ |
|---|---|---|---|---|---|
| R-Tagger | 9.0% | 6.1% | 34.9% | 12.5% | 5.8% |
| R-Tagger' | 42.6% | 39.8% | 55.3% | 34.9% | 6.0% |
| **R-Tagger' AAOD** | **52.4%** | **50.6%** | **80.6%** | **68.9%** | **30.6%** |
| QPIC (Tamura et al., 2021) | 29.9% | | 61.0% | | |
| GEN-VLKT (Liao et al., 2022) | 33.8% | | 64.5% | | |
| HOICLIP (Ning et al., 2023) | 34.7% | | 64.8% | | |
| PViC (Zhang et al., 2023) | 34.7% | | 67.8% | | |
| Pose-Aware (Wu et al., 2024) | 35.9% | | 66.6% | | |
| RLIPv2 (Yuan et al., 2023) | 35.4% | | 68.0% | | |
| **RLIPv2 AAOD** | **36.4%** | | **68.3%** | | |

Table 4: Performance comparisons w/ and w/o the reconstruction module and deformable attention.

| Model | HICO-DET$_b$ testset | | V-COCO testset | | VG$_b'$ valset |
| | $mAP_l$ | $mAP_a$ | $mAP_l$ | $mAP_a$ | $mAP_l$ |
|---|---|---|---|---|---|
| Proposed | 52.43% | 50.55% | 80.57% | 68.94% | 30.58% |
| Control group 1 | 51.75% | 49.35% | 80.81% | 67.37% | 27.49% |
| Control group 2 | 44.09% | 33.62% | 63.59% | 14.82% | 19.43% |
| Control group 3 | 39.14% | 26.82% | 60.81% | 12.49% | 18.12% |
| Control group 4 | 38.85% | 24.19% | 50.54% | 6.34% | 8.92% |
| Control group 5 | 50.27% | 48.32% | 79.52% | 68.42% | 27.04% |

Another issue worth investigating occurs in the second-stage decoder. Since the deformable attention layer relies on corresponding detection boxes to constrain reference points, and the second-stage decoder lacks corresponding supervisory information, the linear layer that maps query outputs to the bounding box refinement vector is essentially frozen. Therefore, it is reasonable to question whether the second-stage decoder converges sufficiently. For comparison, **control group 5** was established in which deformable attention was substituted with classic attention. Since the deformable transformer converges faster, the period before the learning rate decays was extended to 20 epochs, and the total training duration was increased to 50 epochs to ensure sufficient convergence.

For the first hypothesis, as shown in Table 4, when the first-stage decoder is eliminated, the model exhibits substantially inferior performance relative to the original architecture, regardless of whether learnable queries are employed or parameters are compensated. The comparison between control groups 2 and 4 demonstrates that object categories constitute a strong cue that promotes model convergence. The performance of the control group 1 is slightly lower than origin but significantly surpasses that of the control group 3. This suggests that the strategy of buffering first-stage decoder outputs and integrating them into the query inputs across all layers of the second-stage decoder provides considerable benefits for model convergence. For the second concern, control group 5 demonstrate that the classic attention failed to exhibit substantially significant performance gains. Considering computational efficiency, it is reasonable to retain the deformable attention layers.

## 6 CONCLUSION

This paper constructs metrics for the task of human-object interaction inference, and evaluate the proposed query-independent model with cleansed datasets. Results have shown a significant progress compared with the former query-based interaction inference model R-Tagger. Future work includes improving the performance of interaction inference models in zero-shot scenarios and addressing the long-tail distribution challenges prevalent in HOI datasets.

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
