# OpenReview forum: "Decoupled Human-Object Interaction Inference Addressing Architectural Order Dependency in the Query-based Model"
_ICLR.cc/2026/Conference — ICLR 2026 Conference Withdrawn Submission_

### Official Review · Reviewer_csdL · 2025-10-31

**Soundness:** 3
**Presentation:** 3
**Contribution:** 3
**Rating:** 2
**Confidence:** 5

**Summary:**

This paper addresses the core pain point of tight coupling between HOI inference modules and detection pipelines in query-based HOI detection models, aiming to resolve limitations like lack of independent evaluation and optimization constraints caused by coupling. The authors propose a decoupled HOI inference framework: design dedicated evaluation metrics (mAPₗ for labeled pairs, mAPₐ for all pairs); identify two bottlenecks—"architectural order dependency" (decoder self-attention, learnable queries, bidirectional cross-attention) and "dataset impurities" (redundant bounding boxes, verb variants); improve architecture via "removing decoder self-attention + broadcast-style queries" and refine datasets via "IOU-based redundant box clustering + cosine similarity-based verb clustering".

**Strengths:**

•	First to quantify the impact of architectural order dependency from three dimensions and propose targeted improvements; mAPₗ/mAPₐ metrics fill the gap in independent evaluation;
•	Hierarchical experimental design (from dataset to architecture to ablation), cross-benchmark validation (three datasets), and detailed parameters (explicit thresholds/weights);
•	Provides a phased optimization path; open-sourced resources resolve issues of dataset impurities and inconsistent evaluation standards, advancing community research.

**Weaknesses:**

•	Only validated with ResNet-50; no adaptation to mainstream backbones (e.g., Swin Transformer/ViT); no comparison with VLM-based methods (e.g., HOICLIP) to clarify performance boundaries;
•	No cross-validation for VG'; no sensitivity analysis for excluding object-free interactions in V-COCO;
•	No exploration of fundamental solutions for supervision lack in the second-stage decoder; no analysis of impacts on long-tail verbs;

**Questions:**

•	If 5-fold cross-validation or split ratio adjustment is applied to VG', is mAPₗ stable? Can you provide performance fluctuation data?
•	Model Generalization: With Swin Transformer/ViT or RoBERTa-Large, does the 20% mAP improvement remain? Do architectural improvements require parameter adjustments?
•	Method Comparison: Can you supplement comparisons with HOICLIP in zero-shot/long-tail scenarios to clarify applicable boundaries?

---

### Official Review · Reviewer_kR5h · 2025-11-01

**Soundness:** 2
**Presentation:** 2
**Contribution:** 2
**Rating:** 4
**Confidence:** 2

**Summary:**

This paper introduces a decoupled human–object interaction (HOI) inference framework that addresses architectural order dependency in query-based models. By removing self-attention from the decoder and replacing learnable queries with a broadcasted single query, the method eliminates sensitivity to input order. Coupled with dataset refinement strategies such as redundant bounding-box unification and verb clustering, the proposed model significantly improves inference performance.

**Strengths:**

The paper demonstrates strong data-centric contributions by systematically cleaning and unifying noisy annotations: it clusters redundant bounding boxes via IoU-based grouping and adopts a “broadcast” strategy that preserves all relational labels within each cluster, significantly reducing ambiguity in negative sampling; it further normalizes verb diversity across datasets by embedding-based clustering of inflected, phrased and misspelled variants into clean prototypes, which stabilizes text encoder signals and mitigates label imbalance.

**Weaknesses:**

I remain unconvinced that the removal of self-attention layers from the decoder cause input-order insensitive. Perhaps the authors could provide a more explicit clarification.

More fundamentally, I question the strategy of decoupling the interaction inference module from the detection pipeline. Human–object interaction is inherently contextual: the quality of a relation label depends on both the accuracy of the underlying bounding boxes and the global visual context. End-to-end systems can exploit joint supervision and shared representations, and predicts a higher capacity upper bound.

The reported gains appear to derive predominantly from the proposed data-cleansing strategies—redundant-box unification and verb clustering—rather than from the architectural change itself. Perhaps the authors could supply additional comparative experiments to show this point.

**Questions:**

1. Please provide a more detailed explanation of how removing self-attention eliminates the model’s sensitivity to input order

2. Please present additional experimental results that isolate the gains attributable to architectural changes under identical training data—especially a comparison between an end-to-end model trained on the refined dataset and the proposed decoupled architecture.

---

### Official Review · Reviewer_j4LW · 2025-11-04

**Soundness:** 2
**Presentation:** 1
**Contribution:** 2
**Rating:** 2
**Confidence:** 3

**Summary:**

This paper targets to address the key issue in HOI, this is, interaction inference and detection coupled. As it would limit the interaction evaluation and optimization. To address this issue, they proposed a new interaction inference model that removes self-attention from the decoder, which enables the issolation of detection and interaction. Furthermroe, they  identifies two key factors limiting current performance: architectural order dependency and dataset impurities. To address it,. they and introduce dataset refinement strategies, including verb clustering and redundant bounding-box unification. Extensive experiments on multiple benchmarks demonstrates their effectiveness.

**Strengths:**

1. The paper provides comprehensive details on dataset filtering and offers a thorough discussion of architectural order dependency, demonstrating strong analytical depth.

2. It includes extensive experiments and comparisons that effectively support the authors’ claims.

**Weaknesses:**

1. Limited novelty. It is unclear whether this is the first work to decouple detection and interaction prediction. The proposed isolation module appears similar to R-Tagger from RLIPv2. The paper should clarify the differences between the proposed design and R-Tagger, and explain how the new approach specifically addresses the stated optimization issue.

2. Lack of organization and clarity. While the paper presents numerous implementation details, the overall structure lacks coherence, making it difficult to follow the main ideas. The narrative suggests efforts toward dataset cleaning, architectural investigation, and incremental improvements, but it remains unclear what the most effective isolation strategy for interaction and detection actually is.

3. Marginal performance improvement. As shown in Table 3, the performance gains over RLIPv2 are relatively minor (approximately 1% on HICO-DET and 0.3% on V-COCO). It would be helpful to analyze whether these limited gains are primarily due to dataset noise or other factors.

**Questions:**

1. What is the final system design?
2. How would the detection error propagate through the system?
3. Would the interaction estimation serve as context to help reduce the detection error if they are in the same system?

---

### Official Review · Reviewer_SsWd · 2025-11-05

**Soundness:** 3
**Presentation:** 2
**Contribution:** 3
**Rating:** 6
**Confidence:** 2

**Summary:**

This paper proposes a decoupled human–object interaction inference framework that targets a specific weakness in query based detectors, namely their sensitivity to the order and composition of inputs. The authors argue that interaction inference should be evaluated and optimized independently of object detection and association, then diagnose three sources of instability in a popular baseline architecture: self attention in the decoder, learnable queries whose embeddings are tied to input order, and vocabulary sensitivity in language–image fusion. They respond with a redesigned decoder that removes self attention and replaces per pair learnable queries with a broadcasted form, alongside dataset refinements that unify redundant bounding boxes and cluster verbs to reduce label noise. A new evaluation protocol distinguishes performance on labeled pairs from performance over all pairs, explicitly measuring behavior on negatives. Experiments on standard HOI benchmarks report sizable improvements over the decoupled baseline and show modest gains when the refinement is folded back into an end to end detector, supported by ablations that test the role of the reconstruction stage and attention choice.

**Strengths:**

The paper’s strengths are its precise problem formulation and clear motivation for studying interaction inference in isolation, which is timely given the community’s reliance on tightly coupled query based models. The architectural analysis is careful and leads to a simple, implementable change that aligns the model with the stated goal of order invariance while retaining the efficiency advantages of deformable attention. The dataset cleanup is pragmatic and well reasoned, addressing a real source of training ambiguity in HOI corpora and making the evaluation more meaningful. The proposed metrics emphasize behavior on negative pairs, an often underreported aspect of HOI systems with high practical relevance. The experimental narrative is coherent, with qualitative examples and control groups that strengthen the causal link between the diagnosed issues and the reported gains, and the observation that improvements transfer to an end to end detector increases the work’s relevance to practitioners.

---

for rebuttal and discussion, I organized the main strengths as below:

S1. Clear problem framing and motivation for decoupling interaction inference from detection

S2. Simple architectural change that enforces order invariance while preserving efficiency

S3. Practical dataset cleanup that reduces label noise and clarifies evaluation

S4. Coherent experiments and ablations with improvements that transfer to end to end settings

**Weaknesses:**

The main weaknesses are in external validity and evaluation breadth. The approach relies on pre-processing steps such as verb clustering that require threshold tuning and manual verification, which raises questions about reproducibility across datasets and about sensitivity to the chosen encoder for clustering. Treating unlabeled pairs as negatives remains risky in incompletely annotated data, and the paper partly sidesteps this by omitting certain metrics in noisier settings, which makes cross dataset comparisons uneven. The decision to remove self attention trades away explicit modeling of inter pair context; while the goal is order invariance, some scenes may genuinely benefit from interactions among pairs, and the work does not explore alternatives that preserve context without introducing order sensitivity. The claims about estimating an upper bound for end to end detectors would benefit from a clearer theoretical argument or tighter empirical linkage. Finally, the experimental scope could be expanded to include stronger or more diverse language vision baselines, a deeper analysis of computational cost and training efficiency, and a more detailed error analysis on challenging verbs and rare interactions.


---

For rebuttal and discussion:

W1. Dependence on preprocessing such as verb clustering with thresholds and manual checks, raising reproducibility concerns

W2. (minor point) Risky assumption that unlabeled pairs are negatives, leading to uneven cross dataset comparisons

W3. Loss of explicit inter pair context after removing self attention, with limited exploration of alternatives

W4. (important point) Upper bound claims for end to end detectors not convincingly grounded theoretically

W5. Narrow evaluation scope lacking stronger baselines, compute efficiency analysis, and fine grained error breakdowns

**Questions:**

Rather than questions, I have several suggestions. You don’t need to respond to them (they will not affect scores), but please consider them and reply if you have time.

Q1. [permutation-invariant context module] Introduce a permutation invariant context module that recovers cross pair reasoning without reintroducing order sensitivity. After the decoupled predictions are produced, pass human and object tokens through a lightweight graph layer with set pooling to exchange information among nearby pairs, then re score interactions with a residual gating head. This preserves the core design goal while letting true contextual cues like mutual exclusion or role consistency help difficult scenes.

Q2. [counterfactual/compositional training] Augment training with counterfactual and compositional examples to pressure test verb semantics. Programmatically swap humans or objects across images with matched geometry, replace verbs with near opposites using templated captions, and insert hard negatives by cutting and pasting objects into plausible but incorrect contexts. Train with a consistency loss that requires the model to flip or retain predictions appropriately across these counterfactuals.

---

### Official Review · Reviewer_i9Ec · 2025-11-07

**Soundness:** 1
**Presentation:** 1
**Contribution:** 1
**Rating:** 0
**Confidence:** 5

**Summary:**

This paper presents an evaluation framework for isolated HOI inference modules: query-based detectors

**Strengths:**

Refer to weaknesses.

**Weaknesses:**

While the topic addressed by this paper is relevant to the community, the overall quality of the work falls below the standard expected for ICLR. The most significant limitation lies in the lack of a comprehensive and critical survey of related literature, which undermines the claimed novelty and contribution of the paper.

Specifically, the paper overlooks several key lines of research that have already explored the main ideas it presents. For example, The key contribution, which is to individually diagnose object detection with HOI detection has been analyzed in [2]. Prior works have extensively examined the decoupling of object and interaction representations in human–object interaction (HOI) or visual relationship detection tasks [4, 5]. These studies have shown that treating object appearance and relational reasoning as separate but complementary components can lead to more robust and interpretable models. Similarly, the issue of order dependency—how the arrangement or role assignment of entities (e.g., subject–object order) affects relational inference—has also been analyzed in earlier works [4], yet the current paper does not acknowledge these contributions or position its method relative to them.

Furthermore, the use of deformable attention mechanisms to enhance interaction modeling has been previously introduced in [3]. These approaches effectively capture long-range spatial dependencies and flexible relational patterns, and their omission from the discussion makes the proposed method appear less informed by existing advances in the field. Likewise, the idea of unifying multiple object instances into shared interaction regions has been investigated in several previous papers [6,7], aiming to model the joint representation of entities within a shared spatial context. This concept closely resembles the design choices adopted by the current paper, yet without explicit differentiation or justification.

Taken together, the contributions claimed by the authors do not clearly extend beyond these existing works. The absence of a detailed comparison—both conceptually and empirically—makes it difficult to identify any truly novel component in the proposed framework. To meet the expectations of a top-tier venue like ICLR, the paper would need to demonstrate a deeper engagement with prior research, provide clear theoretical or methodological innovation, and present strong empirical evidence showing that the proposed ideas outperform or significantly differ from prior approaches.

In its current form, the paper appears incremental and insufficiently grounded in the existing literature. A more rigorous related-work analysis, coupled with stronger ablation studies highlighting the unique aspects of the approach, would be necessary before the paper could be considered competitive for publication.

[1] DDS: Decoupled Dynamic Scene-Graph Generation Network, WACV'25
[2] Diagnosing Human-Object Interaction Detectors, IJCV'25
[3] MSTR: Multi-Scale Transformer for End-to-End Human-Object Interaction Detection, CVPR'22
[4] Consistency Learning via Decoding Path Augmentation for Transformers in Human Object Interaction Detection, CVPR'22
[5] HOTR: End-to-End Human-Object Interaction Detection with Transformers, CVPR'21
[6] Learning Human-Object Interaction Detection using Interaction Points, CVPR'20
[7] Efficient Adaptive Human-Object Interaction Detection with Concept-guided Memory, ICCV'23

**Questions:**

Refer to weaknesses.

---

### Note · Authors · 2025-11-12

I have read and agree with the venue's withdrawal policy on behalf of myself and my co-authors.